# Dissipation, Residue and Dietary Intake Risk Assessment of Penthiopyrad in Eggplants and Its Removal Using Various Household Processing Techniques

**DOI:** 10.3390/foods11213327

**Published:** 2022-10-23

**Authors:** Li Dou, Shiyin Mu, Guangqian Yang, Jinming Chang, Kankan Zhang

**Affiliations:** State Key Laboratory Breeding Base of Green Pesticide and Agricultural Bioengineering, Key Laboratory of Green Pesticide and Agricultural Bioengineering, Ministry of Education, Guizhou University, Guiyang 550025, China

**Keywords:** penthiopyrad, metabolite, residue, stereoselective, eggplant, processing

## Abstract

A field trial was conducted to illustrate the dissipation and residue and assess the dietary intake risk of penthiopyrad in eggplants, and the distribution was further estimated after different household processing methods. Penthiopyrad dissipated quickly in eggplants, with half-lives of 1.85–2.56 days. The final residue data indicated that following the recommended spraying method, penthiopyrad would not threaten human health. Risk quotient results (<<100%) also demonstrated that the dietary intake risk of penthiopyrad in eggplants for Chinese consumers could be negligible. Washing, peeling and thermal treatments had significant removal effects on penthiopyrad from eggplants (0 < processing factor < 0.60). The characterization of the dissipation and distribution of penthiopyrad in field and processed eggplant samples could provide a more realistic reference for risk assessment of processed products, as well as some information for humans who may be exposed to penthiopyrad.

## 1. Introduction

As we know, applying a high dose of pesticides and not adhering to the stipulated time between the last spray and harvest increases the potential for residues to build up in crops, posing a threat to consumer health [1]. Governments and international organizations regulate the levels of residues in crops for consumption, setting maximum residue limits (MRLs) and pre-harvest intervals (PHIs) [2]. On the other hand, the acceptable daily intake (ADI) and acute reference dose (ARfD) were used to evaluate the long-term and short-term dietary intake risks of pesticides, both of which stipulated the maximum dose that would not cause harm to human health. Relevant studies have shown that the concentrations of pesticides in the growth process of crops decrease with the passage of spraying time, and different processing methods significantly affect the residual levels of pesticides by affecting their physicochemical properties [3]. It is necessary to assess pesticide residues in pre-harvest and processing of crops to ensure the diet safety and health of consumers.

Eggplant (*Solanum melongena* L.) is the third most important crop in the Solanaceae family, and its harvested area and yield are second only to those of tomato and potato [4]. It has high nutritional value and contains not only essential nutrients, such as minerals, vitamins and amino acids, but also active compounds, such as phenolic compounds and pectin [5,6,7]. As a major summer and autumn vegetable, eggplant is cultivated in Northern and Southern China [8]. However, during eggplant growth, some diseases, such as grey mold and verticillium wilt, threaten its cultivation and health. As a result, fungicides need to be applied to protect eggplant and ensure the yield. Penthiopyrad is a carboxamide chiral fungicide that has an excellent curative effect on various plant diseases, such as rust and gray mold [9,10]. PAM, 1-methyl-3-trifluoromethyl-1H-pyrazole-4-carboxamide, is the main plant-derived metabolite of penthiopyrad. The total concentration of penthiopyrad is calculated as the sum of two stereoisomers and PAM. While controlling the disease, the residues of penthiopyrad in eggplants may contaminate the environment and affect human health. At present, the proposed MRL of penthiopyrad set by the European Union (EU) [11] and China [12] is only 2000 μg/kg in eggplant, and the data supporting the safe and appropriate use of penthiopyrad on eggplant are still lacking in China. Our previous study found that the dissipation of penthiopyrad was fast and enantioselective in field-cultivated tomato and cucumber samples, with half-lives of 2.51–7.85 d. The PHI values of penthiopyrad were 1 d and 5 d for cucumbers and tomatoes, respectively, following the recommended application method [13]. The dietary risk results showed that both vegetables were safe when penthiopyrad was used following the suggested application method based on an ADI value of 0.1 mg/kg bw per day and an ARfD value of 1 mg/kg bw [14]. Moreover, few studies have been conducted on the effect of household processing methods on removing penthiopyrad residues from any contaminated vegetables. To ensure food safety and reduce harm to human health, more information about the distribution of penthiopyrad in unprocessed and processed crops is needed to fill the existing gap in China. Therefore, the distribution and potential risk of this fungicide in eggplants should be assessed, and the recommendation of certain efficient household processing methods is essential to removing penthiopyrad residues from contaminated eggplants.

This study used Good Agricultural Practice (GAP) to carry out a trial to investigate the residue behaviors of penthiopyrad and PAM in eggplant under open field conditions with different spray treatments and to assess the dietary risks of penthiopyrad in eggplant based on field data. Moreover, the processing factors (PFs) were calculated and used to compare the effect of general household processing (washing, peeling, steaming and boiling) on the removal of penthiopyrad and PAM residues in eggplant.

## 2. Materials and Methods

### 2.1. Chemicals and Reagents

Pesticide standards, including *rac*-penthiopyrad (purity, 99.5%) and PAM (purity, 99.0%), were bought from J&K Scientific Ltd. (Beijing, China). The formulation of penthiopyrad (20%, suspension concentrate (SC)) was purchased from Mitsui Chemicals (China) Co., Ltd. (Shanghai, China). High-performance liquid chromatography (HPLC)-grade acetonitrile, methanol and formic acid were purchased from Thermo Fisher Scientific (Waltham, MA, USA). Primary secondary amine (PSA) was bought from Agela Technologies (Tianjin, China), and nylon syringe filters were bought from PeakSharp Technologies (Beijing, China). Distilled water was purchased from Watson Group Ltd. (Dongguan, China). All other reagents and solvents used in the experiments were commercially available at analytical purity.

### 2.2. Field Trials

The environmental behaviors of penthiopyrad and PAM in eggplant were studied through field experiments at Huaxi County, Guiyang City, Guizhou Province, during the agricultural season in 2021. Penthiopyrad had not been applied to either of the experimental fields in the past three years. Five treatments were set up in the trials of penthiopyrad in eggplant under field conditions. One plot was given no penthiopyrad treatment, two plots were subjected to 75 g a.i./ha (recommended low dose) two and three times (interval of 7 d) and the other two plots were given 99 g a.i./ha (recommended high dose) two and three times (interval of 7 d). Each plot was 100 m^2^, and between two plots, a 20 m^2^ area was set as a buffer zone. The eggplant samples were randomly collected at 2 h and 1, 2, 3, 5, 7, 10, 14, 18 and 21 d after the last spraying in each treatment to investigate the dissipation kinetics. Among the above sampling times, four intervals (5, 7, 10 and 14 d) were chosen to detect the final residues in eggplant according to the growth characteristics. All samples were homogenized and packed in polyethylene plastic bags, after which they were sealed and labeled properly and stored at −20 °C.

### 2.3. General Household Processing Treatments

The effect of different processing methods on residue reduction was investigated using four common home processing treatments. The eggplants selected require a sufficient primary deposit for subsequent processing [15]. Therefore, in this experiment, fresh eggplant samples of the same size were purchased from the local market. The raw samples were then prepared by soaking eggplants into the solution that contained penthiopyrad (SC/water, 0.5 mL/L) for 5 h. After rinsing off the dipping solution on the surface of the sample with tap water, the sample was naturally dried at room temperature and then placed in black polyethylene plastic bags for 24 h to homogenize. The initial concentration of racemic penthiopyrad in eggplant samples was 4010 ± 220 µg/kg. The samples were divided into five parts, namely contaminated samples without any treatment, peeling (including raw, steamed and boiled eggplants), washing (including physical cleaning and chemical cleaning), steaming and boiling.

### 2.4. Instrumentation and Pretreatment

Based on our previous study [13], the detection of penthiopyrad and PAM was performed on a liquid chromatography with tandem mass spectrometry (LC-MS/MS) system. The extraction of penthiopyrad and PAM from eggplant was conducted with a modified QuEChERS approach. The detailed instrumentation parameters and pretreatment description are listed in Appendix A and the “Extraction and purification” section in Appendix A, respectively. 

### 2.5. Method Validation

The suitability of the established analytical methods was confirmed by the LC-MS/MS chromatogram data of penthiopyrad and PAM in blank, spiked and actual eggplant samples. Linearity, determination coefficient (*R*^2^), limit of detection (LOD), limit of quantification (LOQ), matrix effect (ME), accuracy and precision were used to evaluate the validity of the method [16]. Solvent and matrix-matched standard curves at six concentrations (from 5 to 2000 ng/mL) of penthiopyrad stereoisomers and PAM were drawn to evaluate linearity. The LOD and LOQ were determined with a signal-to-noise ratio (*S/N*) = 3, and *S/N* = 10, respectively [17]. ME values were calculated as the ratio of the slope of matrix-matched standard calibration curve and the slope of solvent standard calibration curve [18]. The intra-day (*n* = 5) and inter-day (*n* = 15) recovery trials were used to evaluated the method’s accuracy and precision. The analytes were spiked in the blank eggplant samples at four concentrations (10, 100, 1000 and 2000 μg/kg), and each test was conducted with five replicates on three consecutive days. The recovery (%) was used to evaluate the accuracy. The precision under experimental conditions was determined by the repeatability and reproducibility of the intra-day and inter-day assays, expressed as relative standard deviation (RSD) [19]. 

### 2.6. Dissipation Kinetics, Processing Factor, Potential Enantioselectivity and Statistical Analysis

The dissipation kinetics of penthiopyrad stereoisomers and PAM in eggplant was evaluated by applying the first-order kinetic equation to the data: *C_t_/C*_0_ = *e*^−*kt*^ and *t*_1/2_ = ln2/*k* [20], where *C*_0_ and *C_t_* are the initial concentration and the concentration (mg/kg) at time (*t*), respectively, *k* is the dissipation rate constant (day^−1^) and *t*_1/2_ is the half-life. The PF based on residue concentrations in the raw crop and the various processed products was calculated as PF = residue concentration in processed product (mg/kg)/residue concentration in raw crop (mg/kg) [21]. The enantiomeric fraction (EF) value was used to evaluate the stereoselectivity of penthiopyrad dissipation in eggplant and calculated as the quantitative ratio of the concentration of *R*-(−)-stereoisomer to the concentration of racemic penthiopyrad [22].

All statistical analyses were performed with SPSS ver. 20.0 software (IBM Corporation, Armonk, NY, USA). One-way analysis of variance and Duncan’s multiple range tests were used to demonstrate the characteristics of each treatment at a confidence level of 0.01. All data are shown as the average value ± standard deviation (SD) from triplicate measurements.

### 2.7. Dietary Risk Assessment

The acute dietary intake risk of penthiopyrad in vegetables for different Chinese consumers was calculated as NESTI = FI×HR/b.w. and RQ_a_ = NESTI/ARfD × 100%. The chronic dietary intake risk was calculated as IEDI = FI × STMR/b.w. and RQ_c_ = IEDI/ADI × 100% [18]. RQ_a_ and RQ_c_ are the acute risk quotient and chronic risk quotient, respectively. According to risk assessment calculation, RQ > 100% indicates the presence of adverse effects of pesticide dietary intake on consumer health; on the contrary, RQ < 100% indicates no significant health risk for consumers [23]. FI is the food consumption intake gathered from the Chinese intake survey report (g/d) [24], NESTI is the national estimated short-term intake (mg/kg b.w.), HR is the highest residue level (mg/kg), b.w. is the body weight (kg), ARfD is the guidance value for daily exposure from short term intake (mg/kg b.w.), IEDI is the international estimated daily intake (mg/kg b.w.), STMR is the supervised trial median residue (mg/kg) and ADI is the acceptable daily intake (mg/kg b.w.) [25].

## 3. Results

### 3.1. Method Validation

Solvent and matrix-matched calibration standard curves were constructed using a series of analyte concentrations plotted against chromatographic peak areas. With the developed pretreatment and detection procedures, the LC-MS/MS chromatograms of penthiopyrad stereoisomers and PAM in standard solution, blank, spiked and actual eggplant samples are presented in Appendix A. The regression equations and *R*^2^ data of the standard solution and matrix-matched curves (Table 1) showed that good linearity was achieved for penthiopyrad stereoisomers and PAM (*R*^2^ > 0.99 in all cases). The ME results showed that the signals of the two penthiopyrad stereoisomers were slightly enhanced by the eggplant matrices due to the ME values of 1–1.04. The LODs of all analytes were 3.3 µg/kg, and the respective LOQs were 10 µg/kg in eggplant. The relevant parameters for method validation of penthiopyrad stereoisomers and PAM in eggplant are listed in Table 2. The intra-day recoveries of *R*-(−)-stereoisomer and *S*-(+)-stereoisomer were 72.1–113.8%, and the intra-day RSDs were 0.4–10.4%. The inter-day recoveries of two stereoisomers were 78.9–101.2% with the inter-day RSDs ≤ 7%. The intra-day and inter-day recoveries for PAM were 79.1–97.7% and 82.2–94.4%, respectively, with their RSDs below 9%.

### 3.2. Dissipation, Residue Distribution and Dietary Risk Assessment of Penthiopyrad and PAM in Eggplants under Field Conditions and the Potential Stereoselectivity

The quality control (QC) of actual eggplant sample detection is shown in Appendix A. The mean recoveries of penthiopyrad stereoisomers and PAM in eggplant samples were in the range of 84.6% to 91.2%, and the RSDs were between 1.7% and 4.6%, indicating that the detection method of penthiopyrad and PAM is stable and the detection of the field eggplant samples is reliable. The corresponding parameters for the dissipation dynamics trial are listed in Table 3. In four treatments, penthiopyrad stereoisomers dissipated more than 99% in eggplant samples gathered at 14 d after the final spraying. In four application treatments, after 2 h of application, the residual concentrations of racemic penthiopyrad in the eggplant were 118, 158, 339 and 340 µg/kg. Subsequently, levels of penthiopyrad stereoisomers were gradually reduced over time. After 21 days of application, the residual concentrations of racemic penthiopyrad were all <1.4 µg/kg, much lower than the MRLs. The half-lives of the *R*-(−)- and *S*-(+)-stereoisomers were 1.85–2.49 and 1.88–2.56 days, respectively (Duncan’s multiple range test, *P* < 0.001, Appendix A). Meanwhile, the metabolite PAM was found during the dissipation trial. The concentrations of PAM initially rose and then declined with the increase in sampling time in the four experimental treatments (Appendix A). As shown in Figure 1, the EF values of penthiopyrad stereoisomers did not deviate significantly from 0.5 (Appendix A).

The final residues of penthiopyrad stereoisomers, racemate and total penthiopyrad (level of *rac*-penthiopyrad + 1.86 × level of PAM) in eggplant are shown in Appendix A. In eggplant samples cultivated in field condition, the residue concentrations of *R*-(−)-stereoisomer were <10–48 µg/kg, those of *S*-(+)-stereoisomer were <10–50 µg/kg, those of *rac*-penthiopyrad were <10–98 µg/kg, those of PAM were < 10 µg/kg and those of total penthiopyrad were <10–102 µg/kg (*P* < 0.001) at 5, 7, 10 and 14 d after the last spray in four treatments. According to the survey report on the nutrition and health of Chinese residents [22], the body weights were 12.3–64.9 kg and the daily intakes were 39.6–99.5 g/d for different age and gender groups of Chinese consumers. The STMRs and HRs of penthiopyrad stereoisomers, *rac*-penthiopyrad and total penthiopyrad, in eggplant were 1–46 and 3–111 µg/kg, respectively. The ARfD (1 mg/kg b.w.) and ADI (0.1 mg/kg b.w.) of penthiopyrad were chosen [14]. The RQ_a_ and RQ_c_ values of penthiopyrad stereoisomers in Figure 2, racemate and total penthiopyrad, were calculated as <0.1% (Appendix A) and <0.3% (Appendix A), respectively.

### 3.3. The Distribution of Penthiopyrad Residues in Eggplant Samples after Household Processing

#### 3.3.1. Washing Processing

In this experiment, physical washing and chemical washing were adopted. The results showed that all washing with the aqueous media reduced the concentration of penthiopyrad in the tested eggplants. The effect of washing treatments on residues of penthiopyrad in eggplant is summarized in Table 4. The reduction ratios (%) are shown in Appendix A. The PFs and reduction ratios were in the range of 0 to 0.59 and 41% to 100%, respectively. In the eggplants with physical washing, the decreased concentration levels of penthiopyrad stereoisomers were in the range of 63 to 94% (PF = 0.36–0.06). The reduction effect corresponding to the three cleaning methods is tap water rinsing (W1) > stirring and soaking (W2) > static and soaking (W3). In chemical washing, the aqueous solutions of sodium bicarbonate (W4), acetic acid (W5), sodium chloride (W6), ethanol (W7) and sodium dodecyl benzene sulfonate (W8) were used to evaluate the effectiveness of washing eggplants in reducing penthiopyrad residues. Here, sodium bicarbonate, acetic acid, sodium chloride, ethanol and sodium dodecylbenzene sulfonate were used to replace the baking soda, vinegar, salt, wine and some vegetable and fruit washing agents commonly used in ordinary households, respectively. These chemicals are added to water to increase the solubility of pesticides in water. As shown in Appendix A, the reduction ratios were in the range of 57% to 100% (PF = 0.43–0); 80% of the reduction ratios were above 90%. Among them, the reduction effect of W4, W6, W7 and W8 is similar, while that of W5 is slightly lower.

#### 3.3.2. Peeling Processing

As shown in Table 4 and Appendix A, when the raw, steamed and boiled eggplant samples were peeled separately, the removal rates of penthiopyrad were all approximately 100% (PF ≈ 0). At the same time, the ratio of peel and pulp significantly affected the distribution of penthiopyrad in eggplant. In eggplant, the water content of eggplant peel is much lower than that of eggplant pulp. Penthiopyrad is a low-water-soluble (7.53 mg/L, 20 °C) fungicide which diffuses slowly in the eggplant pulp. Therefore, penthiopyrad is mainly deposited on the eggplant peel, which can be easily and completely removed by peeling.

#### 3.3.3. Thermal Processing

In the experiment, the contaminated eggplant samples were added when the water boiled, and the effect of steaming and boiling at different times on the elimination of penthiopyrad in the samples was studied. The QC of actual steamed and boiled eggplant sample detection is shown in Appendix A. The average recoveries of penthiopyrad stereoisomers and PAM in steamed and boiled eggplant samples were from 84.6% to 92.4% with RSDs of 1.5–4.2%. The data demonstrate the adequate stability of the penthiopyrad and PAM detection method, as well as the acceptable reliability of the detection of actual steamed and boiled eggplant samples. Compared with water treatment at room temperature, steaming treatment with boiling to 100 °C in an open system helped to reduce the levels of penthiopyrad in eggplants. However, during the steaming process, the skin of the eggplant may be partially damaged, making it easier for contaminants in the water to diffuse into the pulp, resulting in an unsatisfactory reduction effect. During the water boiling point thermal processing, steaming for 20 min had the best reduction ratio of 99% (PF = 0.01, Table 4). As shown in Appendix A, it was observed that the anthocyanin content in eggplant samples decreased significantly as a result of thermal processing treatment. The color of eggplant peels in non-thermal processing is dark purple, and the dark purple is slightly lighter in boiling and tawny in steaming. Obviously, eggplant peels behave differently in the face of different thermal treatments. The destruction of the eggplant peels during the boiling treatment allowed the penetration of penthiopyrad in the surrounding solution into the eggplant interior. However, in the steaming treatment, although the thermal treatment also destroyed the stability of the substances in the eggplant peel, the eggplant peel was not in contact with the boiling water containing penthiopyrad but was repeatedly steamed and washed with steam. Thus, although the temperature and treatment time are the same, the corresponding reduction ratios are different.

## 4. Discussion

Good linearity and low spiked level meet the method performance criteria for recovery and precision, and average recovery for each spiked level tested in the range of 70 to 120% and RSD ≤ 20% were used to demonstrate that the analytical method was feasible [26]. Some components presented in the matrix might influence the analytical signal of the pesticide, resulting in a differential recovery rate [18]. All of the data in the method validation trials confirmed that the established analytical approach followed the requirements of the CXG 90-2017 guidelines [27]. The dissipation of penthiopyrad stereoisomers all followed the first-order kinetic model with correlation coefficient (*r*^2^) > 0.94 [28]. Penthiopyrad dissipated fast (*t*_1/2_ < 3 d), and the residual concentrations of penthiopyrad were all lower than the temporary MRLs of China (2000 µg/kg). The changes (decrease after increase) of PAM levels during the dissipation process possibly contributed to complicated metabolism factors in vegetables [29]. Although the dissipation of the *R*-(−)-stereoisomer was found to be a little more than that of the *S*-(+)-stereoisomer at 21 days, since the amount of penthiopyrad present in the eggplant at this time was quite low, the probability of errors in the data also increased accordingly. Moreover, because an EF value of 0.5 indicated no stereoselective dissipation [18], the penthiopyrad stereoisomers were approximately free of stereoselective degradation in eggplants. Overall, under field conditions, there was no stereoselective degradation of racemic penthiopyrad in eggplants, and its rate of dissipation was less different among the four application treatments, suggesting that spray dose and time may not affect penthiopyrad stereoisomer-selective dissipation in eggplants. However, in our previous study, penthiopyrad stereoisomers dissipated differently in cucumber (or tomato) samples under open field or greenhouse conditions [13]. The difference may be related to the specific physicochemical properties of different vegetables and the complex microbial communities in the cultivation ecosystem [28], which can be the priority and direction of our future research.

Racemic penthiopyrad was sprayed on the eggplants in order to obtain a proper penthiopyrad-based safety assessment regime. In this study, pesticide concentrations were measured at various stages of eggplant growth. The pre-harvest interval (PHI) could be recommended based on the comparison of MRL and final residue data [30]. As shown in Appendix A, the final residues of penthiopyrad in field cultivated eggplant samples were all lower than 2000 µg/kg (Chinese temporary MRL in solanaceous vegetable). The data demonstrate that after spraying penthiopyrad using the recommended approach, the PHI in the eggplant is 5 d, and the field-cultivated eggplant samples may be safe for Chinese consumers. Pesticide residues primarily migrate to humans through consumption of contaminated products. Besides the MRL evaluation for final residue of penthiopyrad in eggplant, dietary intake risk assessment approach was further conducted to estimate the safety of penthiopyrad in eggplant [30]. In Figure 2, all RQ_a_ values and more than 80% of RQ_c_ values were less than 0.1%, which further indicated that following the recommended application method, the dietary intake risk of penthiopyrad could be negligible in field-cultivated eggplants for Chinese consumers [25].

PFs are commonly used to evaluate the impact of household processing on pesticide residues and play an important role in assessing the health risks. According to the relevant regulations of the European Food Safety Authority [21], a PF value < 1 (or >1) indicates a decrease (or an increase) in residual pesticide levels after processing, while a PF value of 1 shows no changes of residual pesticide levels in processed agricultural products [31]. Whether the eggplants are picked from the field or bought from the market, washing is the primary method of processing and an initial step in general household processing. Multiple studies have proved that washing could effectively reduce pesticide residues in vegetables [32]. In the present study, the results show that a large amount of penthiopyrad residue can be removed by washing with tap water. One possible reason is that the pesticides are washed away directly by the flow of the tap water, rather than being stored in a stationary container [33]. Although Miyahara and Saito mentioned that mechanical agitation can also increase the reduction ratio of pesticide residues in crops [34], the removal efficiency of instrument-assisted soaking for penthiopyrad in eggplant samples was still not as good as that of tap water rinsing. In general, soaking vegetables with a certain amount of chemicals, such as acid, alkali or salt, is more effective at removing pesticides than water alone [35,36]. However, ordinary household detergents often contain other contaminants, which is another concern. As a result, chemical washing is not recommended unless necessary in comparison to tap water washing. Peeling is another important step in the general household processing of eggplants. Previous studies have reported that peeling not only removes most pesticides from agricultural products but can even completely remove them. Unprocessed grapes had the highest concentrations of cyanobacterial and the main metabolite, CCIM, and peeling reduced them by 95.0% and 78.0%, respectively [35]. In tomatoes, also a commercial crop of the Solanaceae family, washing and peeling effectively removed almost all chlorpyrifos and ethyl-parathion [1]. A similar study found that acetamiprid and permethrin residues were undetectable in peeled tomato samples. The peeling process of tomatoes effectively removes almost all fungicide residues [15]. Obviously, peeling can completely remove penthiopyrad from eggplants. This is expected since penthiopyrad has a non/low octanol/water coefficient (log*K_ow_* = 4.62), which allows it to be immobilized in plant tissue [1]. Different thermal processing treatments may have different reduction ratios of pesticides. Thermal processing treatments, including steaming and boiling, have been shown to be effective in breaking down various pesticides. In thermal processing, the processing conditions, such as temperature, duration and presence or absence of fluid and certain physicochemical properties of pesticides, have an impact on the removal of pesticide residues. As the temperature increases, the pesticide volatilizes, hydrolyzes or thermally degrades, resulting in a decrease in its concentration [37]. From the results of steaming and boiling, lower concentrations of penthiopyrad were obtained compared to the initial data, regardless of whether there was liquid contact during processing, and the determined PF was <1. Eggplant peel contains many phenolic compounds, pectin and dietary fiber and other substances, which give eggplants a defense against pesticide pollution. The strong anthocyanins (the main phenolic substances in eggplant peel) accumulate and make the eggplant peel dark purple [38]. Thermal processing treatment has a noticeable effect on the stability of eggplant phenols, especially anthocyanins. Over-ripening reduces the content of anthocyanins, making the dark purple color lighter, eventually resulting in a brown product [39]. Related research shows that boiling is the best way to retain anthocyanins during heat treatment [5]. Martini et al. found that eggplants lost the most after roasting (91.2% loss), followed by grilling (84.4% loss), frying (60.3% loss) and boiling (43.7% loss) [40]. In a study of separate and mixed thermal treatments containing dietary fiber and anthocyanins, it was found that one of the possible reasons for the retention of anthocyanins was the encapsulation of anthocyanins in the fiber matrix, and the other was the interaction between the dietary fiber and cyanine. This micro-interaction between anthocyanins and dietary fiber may contribute to their thermostability [5]. The reduction ratio of penthiopyrad in the boiling treatment was significantly lower than in the other treatments, possibly related to the stability that the thermal treatment may destroy. This experiment investigated the effect of washing, peeling, steaming and boiling on the penthiopyrad residues. All of these values are <1, illustrating that these processes facilitate pesticide removal and reduce exposure risk. The results showed that PF of peeling was significantly lower than that of the other processes, and peeling was the most effective method of removing penthiopyrad from eggplant. For consumers, after washing, agricultural products should be peeled as much as possible before subsequent processing to minimize pesticide residues. At the same time, the penthiopyrad metabolite PAM was not found during these procedures. This may be related to the contamination time of penthiopyrad. At present, data on the metabolite concentrations of penthiopyrad in processed commodities are still lacking. In future studies, it is necessary to investigate the formation of pesticide metabolites in processed products to further ensure food safety.

## 5. Conclusions

In this study, the residual levels of penthiopyrad and PAM in eggplants were determined by LC-MS/MS. The reliability of the established analytical methods was demonstrated by linearity, ME, recovery, LOD, LOQ, accuracy and precision. The dissipation behavior of penthiopyrad in the field-cultivated eggplant samples is consistent with the first-order kinetic dissipation regulation, and the metabolite PAM was detected. The results showed that the content of racemic penthiopyrad and its metabolite PAM in eggplant at 2 h and 21 d were lower than all current MRLs. Both RQ_a_ and RQ_c_ are below 1%, indicating that the field-cultivated eggplant samples are safe for different Chinese consumer groups according to the recommended application method of penthiopyrad. The effects of washing, peeling and thermal treatment on the residual levels of penthiopyrad in the eggplants were also investigated. All of these treatments reduced the excess penthiopyrad residues in eggplants to below the MRL, indicating that general household processing of the eggplants was sufficient to reduce the risk of penthiopyrad residues. Among them, peeling was the most effective approach to remove penthiopyrad from eggplant (PF = 0), followed by steaming in thermal treatment (PF < 0.04). The metabolite PAM was not found during processing. Neither field nor processing experiments showed that there was no stereoselectivity for the dissipation of penthiopyrad stereoisomers. In summary, the present work has found that neither the residual levels of penthiopyrad during the growing process of eggplant nor the residual levels after processing pose a health risk to Chinese consumers of different ages. The results may serve as a reference for farmers on the safe application of penthiopyrad in vegetables and other crops and as a more realistic assessment for producers and processors monitoring pesticide residue levels to improve food safety.

## Figures and Tables

**Figure 1 foods-11-03327-f001:**
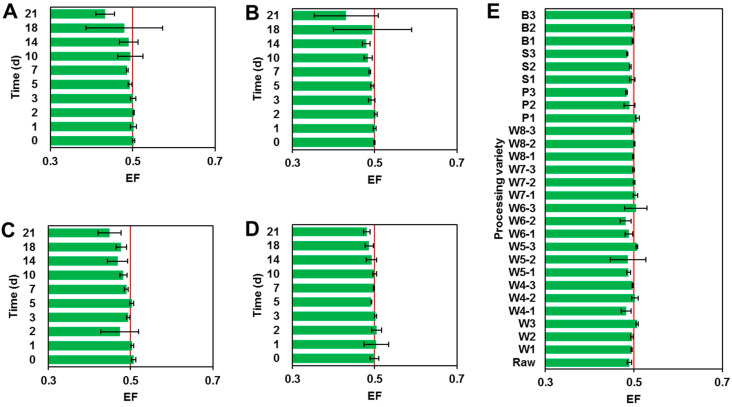
EF values of penthiopyrad in eggplant samples under field conditions. (**A**–**E**) represent spraying dose of 75 g a.i./ha two and three times, spraying dose of 99 g a.i./ha two and three times and processing, respectively.

**Figure 2 foods-11-03327-f002:**
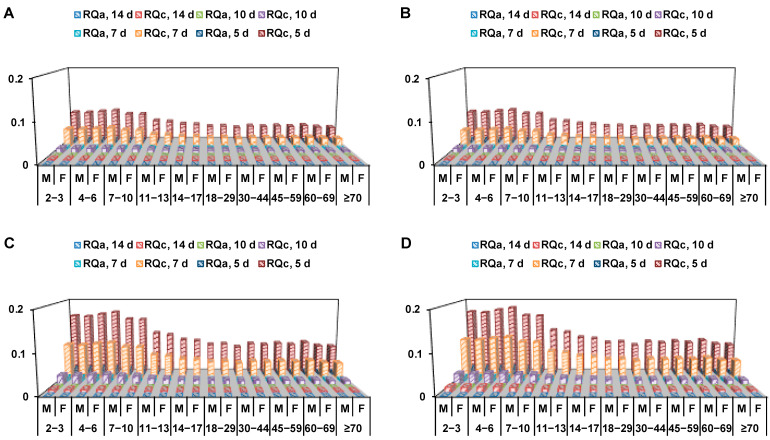
Visual diagram of acute and chronic risk quotient values of *R*-(−)-stereoisomer (**A**), *S*-(+)-stereoisomer (**B**), *rac*-penthiopyrad (**C**) and total (sum of racemic penthiopyrad and PAM) penthiopyrad (**D**) in field-cultivated eggplant samples gathered from different treatments for different groups of Chinese consumers (M: Male and F: Female).

**Table 1 foods-11-03327-t001:** Linear equation, determination coefficient (R2), LOD, LOQ and matrix effect (ME) of penthiopyrad stereoisomers and PAM in solvent and eggplant samples.

Matrix	Analyte	Regression Equation	*R* ^2^	ME	LOD (µg/kg)	LOQ (µg/kg)
Acetonitrile	*R*-(−)-stereoisomer	*y* = 417,350*x* + 18,241	0.9985	/	/	/
	*S*-(+)-stereoisomer	*y* = 420,162*x* + 17,019	0.9989	/	/	/
	PAM	*y* = 2,000,000*x* + 136,295	0.9959	/	/	/
Eggplant	*R*-(−)-stereoisomer	*y* = 434,925*x* + 19,769	0.9980	1.04	3.3	10
	*S*-(+)-stereoisomer	*y* = 430,125*x* + 21,207	0.9977	1.02	3.3	10
	PAM	*y* = 2,000,000*x* + 127,874	0.9960	1.00	3.3	10

**Table 2 foods-11-03327-t002:** Recoveries and relative standard deviations (RSDs) of penthiopyrad stereoisomers and PAM in eggplant.

Analyte	Matrix	Spiked Level (µg/kg)	Intra-Day Recovery, RSD (%, *n* = 5)			Inter-Day Recovery, RSD (%, *n* = 15)
			Day 1	Day 2	Day 3	
*R*	Eggplant/raw	10	81.8, 7.6	105.7, 8.4	113.8, 4.9	100.4, 7.0
		100	86.6, 1.5	89.4, 6.5	86.3, 3.8	87.5, 3.9
		1000	98.7, 1.2	90.5, 5.1	97.7, 1.9	95.6, 2.7
		2000	85.3, 0.6	87.1, 3.1	82.0, 1.7	84.8, 1.8
*S*	Eggplant/raw	10	81.9, 8.3	97.2, 7.7	101.1, 3.5	93.4, 6.5
		100	85.6, 0.6	84.5, 4.2	88.6, 3.2	86.2, 2.7
		1000	96.4, 2.3	87.7, 4.8	93.7, 1.2	92.6, 2.8
		2000	86.7, 1.9	84.1, 2.5	82.6, 2.2	84.5, 2.2
PAM	Eggplant/raw	10	85.4, 4.5	82.2, 3.0	79.1, 3.9	82.2, 3.8
		100	87.5, 2.3	86.6, 2.2	85.4, 2.1	86.5, 2.2
		1000	88.4, 2.7	83.4, 1.8	81.7, 1.4	84.5, 1.9
		2000	91.4, 3.0	87.6, 0.9	86.6, 1.6	88.5, 1.8
*R*	Eggplant/steamed	10	89.0, 1.7	106.8, 3.7	107.8, 2.5	101.2, 2.6
		100	93.0, 5.1	91.6, 3.2	90.4, 2.5	91.7, 3.6
		1000	73.8, 2.0	85.0, 2.5	84.2, 1.9	81.0, 2.1
		2000	79.2, 1.6	89.8, 2.3	87.2, 2.1	85.4, 2.0
*S*	Eggplant/steamed	10	85.3, 1.2	94.8, 3.4	93.6, 6.9	91.2, 3.9
		100	76.7, 1.7	92.6, 1.7	92.4, 2.6	87.2, 2.0
		1000	72.5, 1.9	84.7, 4.6	82.1, 2.3	79.8, 2.9
		2000	79.8, 1.7	89.8, 2.8	87.0, 1.0	85.6, 1.8
PAM	Eggplant/steamed	10	91.1, 8.8	87.4, 7.6	84.0, 7.2	87.5, 7.8
		100	86.1, 1.8	86.8, 6.7	87.0, 6.8	86.7, 5.1
		1000	92.6, 7.1	88.7, 7.1	87.1, 8.4	89.5, 7.6
		2000	91.9, 3.9	87.8, 3.9	89.0, 3.8	89.6, 3.9
*R*	Eggplant/boiled	10	94.2, 4.1	95.6, 8.1	96.1, 7.2	95.3, 6.4
		100	93.3, 4.5	91.4, 3.7	94.5, 3.7	93.0, 4.0
		1000	77.6, 8.2	84.1, 2.9	81.3, 1.8	81.0, 4.3
		2000	78.7, 0.4	91.7, 10.4	82.1, 10.3	84.2, 7.0
*S*	Eggplant/boiled	10	90.4, 3.6	88.3, 4.2	90.1, 6.2	89.6, 4.7
		100	77.1, 0.9	95.5, 2.0	90.2, 1.6	87.6, 1.5
		1000	72.1, 1.9	84.1, 3.3	80.5, 2.3	78.9, 2.5
		2000	79.5, 0.7	90.2, 4.4	86.9, 0.7	85.5, 1.9
PAM	Eggplant/boiled	10	85.8, 4.7	85.8, 5.2	90.6, 6.9	87.4, 5.6
		100	86.8, 5.9	89.9, 4.2	93.2, 4.5	90.0, 4.9
		1000	97.7, 8.6	93.6, 9.0	92.0, 7.8	94.4, 8.5
		2000	96.4, 4.1	91.4, 4.6	91.5, 4.7	93.1, 4.5

**Table 3 foods-11-03327-t003:** Dissipation parameters of two penthiopyrad stereoisomers in field-cultivated eggplants.

Dose (g a.i./ha)	Spraying Time	Stereoisomer	*r* ^2^	*P* _1_	*k* (d^−1^)	*P* _2_	*t*_1/2_ (d)	*P* _3_
75	2	*R*	0.9551 ± 0.0038 ^c^	<0.001	0.2806 ± 0.0040 ^b^	<0.001	2.47 ± 0.04 ^a^	<0.001
		*S*	0.9476 ± 0.0042 ^c^		0.2712 ± 0.0117 ^b^		2.56 ± 0.11 ^a^	
	3	*R*	0.9654 ± 0.0107 ^b^		0.3772 ± 0.0325 ^a^		1.85 ± 0.16 ^b^	
		*S*	0.9648 ± 0.0067 ^b^		0.3686 ± 0.0178 ^a^		1.88 ± 0.09 ^b^	
99	2	*R*	0.9871 ± 0.0025 ^a^		0.3665 ± 0.0198 ^a^		1.89 ± 0.10 ^b^	
		*S*	0.9866 ± 0.0021 ^a^		0.3581 ± 0.0204 ^a^		1.94 ± 0.11 ^b^	
	3	*R*	0.9842 ± 0.0038 ^a^		0.2789 ± 0.0069 ^b^		2.49 ± 0.06 ^a^	
		*S*	0.9832 ± 0.0037 ^a^		0.2751 ± 0.0050 ^b^		2.52 ± 0.05 ^a^	

Data are expressed as average values ± SD. *P*_1_, *P*_2_ and *P*_3_ represent confidence within the 99% confidence level of correlation coefficients (*r*^2^), first-order rate constants (*k*) and half-lives (*t*_1/2_), respectively; different lowercase letters indicate statistical significance between different treatments for field-cultivated eggplant according to Duncan’s multiple range test (*P* = 0.01).

**Table 4 foods-11-03327-t004:** PFs of penthiopyrad in eggplant samples after different processing methods.

Processing Variety		Number	Concentration	Time/min	PF (*R*)	PF (*S*)	PF (*Rac*)
Washing	Tap water rinsing	W1		30	0.07	0.06	0.06
	Stir and soaking	W2		30	0.25	0.25	0.25
	Static and soaking	W3		30	0.38	0.35	0.36
	Sodium bicarbonate	W4	1%	30	0.01	0.01	0.01
			0.5%		0.02	0.02	0.02
			0.2%		0.15	0.15	0.15
	Acetic acid	W5	1%	30	0.03	0.03	0.03
			0.5%		0.09	0.09	0.09
			0.2%		0.45	0.42	0.43
	Sodium chloride	W6	1%	30	0.02	0.02	0.02
			0.5%		0.07	0.07	0.07
			0.2%		0.11	0.10	0.10
	Ethanol	W7	1%	30	0.01	0.01	0.01
			0.5%		0.04	0.04	0.04
			0.2%		0.13	0.12	0.12
	Sodium dodecyl benzene sulfonate	W8	1%	30	0	0	0
			0.5%		0.06	0.06	0.06
			0.2%		0.09	0.09	0.09
Peeling	Raw	P1			0	0	0
	Steaming	P2		10	0	0	0
	Boiling	P3		10	0	0	0
Steaming		S1		5	0.04	0.04	0.04
		S2		10	0.01	0.01	0.01
		S3		20	0.01	0.01	0.01
Boiling		B1		5	0.60	0.58	0.59
		B2		10	0.51	0.50	0.51
		B3		20	0.30	0.29	0.30

## Data Availability

Data are contained within the article and Appendix A.

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
