# Peer review of "Dissipation, Residue and Dietary Intake Risk Assessment of Penthiopyrad in Eggplants and Its Removal Using Various Household Processing Techniques"

_foods, 2022, doi:10.3390/foods11213327_

Round 1

Reviewer 1 Report

The paper shows the results of penthiopyrad residues in eggplant. The pesticide is registered in various countries, including USA, Canada, Australia, and in Europe, and is widely used in these countries. In the pesticide area, a residue study conducted with only one pesticide and one crop has little relevance, both from the analytical and dietary risk assessment. The analytical methods should be multiresidues, to be used in monitoring programs. Furthermore, the analytical method used in the study was already validated previously for similar crops (Yang et al., 2022), so, nothing new in this regard.

                The authors talk about residues being safe because they are below the MRL. However, MRL is not a safety parameter, but reflect the maximum residue level expected when the product is applied in the field according to the good agricultural practice (GAP), which are is the use instruction of the product in a given crop (concentration of the active ingredient, number of application, interval between the applications and time between the last application and harvest. Currently, there is no registration for penthiopyrad in eggplant in China, and, although the authors does not specifically say this, it seems that the data generated here are to support a recommendation of 2 mg/kg. The authors seem not to be familiar with dietary intake assessment principles

Other specific points are outlined below:

Abstract: penthiopyrad is already widely used in agriculture, and there is no lack of data. The pesticide was evaluated by the FAO/WHO JMPR in 2012, where residue data on various crops was evaluated, including declining and processing studies. The monograph can be found at  (https://www.fao.org/3/cb2755en/cb2755en.pdf).

Introduction. The text on lines 48 to 61 has no relevance for the subject. Lines 63-75: field trials are conducted to generate data for the establishment of MRL. If a MRL has already being proposed in China, the study is useless. kind of studies– those are residue definitions, which are different for enforcement and dietary intake. Reference 11 does not support the last sentence.

Materials and methods: line 104  - it should be ha, not hm2. Is this the GAP in China?              Line 114 – this is not the correct way of conducting a processing study, as the product is not used post-harvest. The authors should have used samples from the field trials with incurred residues. What is chemical and physical washing?

Line 137 – this method was already reported and validated previously (Yang et al., 2022). Line 166-177 – this should be a different section. The concept for dietary risk assessments are wrong. First of all, there is no relevance on estimate a chronic intake from the consumption of a single commodity. Secondly, FI was taken from the WHO Cluster diet, which is a consumption level for chronic exposure and should not be used for acute exposure. Latter in the text (line 233) the authors mention a different consumption value, specific for the Chinese population

Results: Table 1 can go to supplementary material. Again, the method is not new and was already validated for similar crops by the same group

Table 2. Dissipation of the stereoisomers of penthiopyrad was already reported for similar crops by the same group (Yang et al.,2022), nothing new here again.

Lines 209-222 – dissipation of residues over time are expected. Which MRLs the authors are talking about here? There is a single MRL for a given pesticide/crop, which is related to a single post-harvest interval.

Discussion – the whole discussion about safety related to MRL does not make sense, and all conclusions reached by the field and processing studies are expected – reduction of the pesticides.

Reviewer 2 Report

The authors presented a manuscript titled " Residue Behaviour, Risk Evaluation and Household Processing Effect of Chiral Fungicide Penthiopyrad in Eggplant" in order to  illustrate the residual behavior of penthiopyrad, and the distribution was estimated after different processing methods. However, on lines 81-84 the authors provided three aims, which is not in line with the abstract. 

Reviewer 3 Report

The paper is interesting. Investigation of residue behavior, risk evaluation, and household processing effect of Penthiopyrad in eggplant can be important topics from a risk assessment point of view. Anyway, there are a lot of limitations in the study it is important to have precise data. 

Major comments:

1. From a dietary exposure point of view investigations that include only one agricultural food commodity must explain the selection of that product and contribution to the diet because HBGV's are established for the chemical substance, which can have different sources, and obviously aggregate risk assessment is needed in order to understand public health risks. Due to the fact, that investigation is not applying the method of MOE (margin of exposure) which can provide also exposure data from a single food commodity the selection of the product both from chemical occurrence and dietary exposure point of view must be justified.    

2. The study is not involving dietary intake estimation of eggplant for different cooking methods. There is no also study of eggplant consumption patterns. It is mentioned that FI is the food consumption intake 173 gathered from the Global Environmental Monitoring System/Food cluster diets (g/d). No reference. No explanation why GEMS was used as a source of food consumption. Having a huge experience in food consumption data collection I don't think that eggplant is used on a daily basis. So detailed description of consumption patterns, seasonal variation etc are needed. 

3. Extensive revision of the introduction is needed because it is not highly correlated with the main outcomes of the research.  

4. The study is not providing a worst-case scenario, because there is no survey of eggplant consumption and cluster analysis which is also a limitation of the study. 

5. Quality control and quality assurance of chemical analysis must be provided.

6. Conclusion that the findings may serve as recommendations for supplementary review of penthiopyrad MRLs and corresponding processing factors for supplementary data is not justified. Worst-case scenarios are missing. I don't consider that the precision and limitations of the study are allowing to have such kind of conclusion.

Round 2

Reviewer 3 Report

Thanks a lot for taking into account my recommendations. Although I do believe that the paper needs more precise data in order to assess health risks, the authors made an effort and extensively changed the paper. So my suggestions can be used for future research. 

Author Response

Thanks for your recommendation.